# Hydrogel Formulations of Antibacterial Pyrazoles Using a Synthesized Polystyrene-Based Cationic Resin as a Gelling Agent

**DOI:** 10.3390/ijms24021109

**Published:** 2023-01-06

**Authors:** Silvana Alfei, Guendalina Zuccari, Eleonora Russo, Carla Villa, Chiara Brullo

**Affiliations:** 1Department of Pharmacy, Section of Chemistry and Pharmaceutical and Food Technologies, University of Genoa, Viale Cembrano, 4, 16148 Genoa, Italy; 2Department of Pharmacy (DIFAR), Section of Medicinal Chemistry and Cosmetic Product, University of Genoa, Viale Benedetto XV, 3, 16132 Genoa, Italy

**Keywords:** antibacterial pyrazoles, cationic polystyrene-based resin (R1), self-formed composite hydrogels, high-level porosity, excellent swelling capability, pseudoplastic rheological behavior

## Abstract

Here, to develop new topical antibacterial formulations to treat staphylococcal infections, two pyrazoles (**3c** and **4b**) previously reported as antibacterial agents, especially against staphylococci, were formulated as hydrogels (R1-HG-3c and R1HG-4b) using a cationic polystyrene-based resin (R1) and here synthetized and characterized as gelling agents. Thanks to the high hydrophilicity, high-level porosity, and excellent swelling capabilities of R1, R1HG-3c and R1HG-4b were achieved with an equilibrium degree of swelling (EDS) of 765% (R1HG-3c) and 675% (R1HG-4b) and equilibrium water content (EWC) of 88% and 87%, respectively. The chemical structure of soaked and dried gels was investigated by PCA-assisted attenuated total reflectance (ATR) Fourier transform infrared (FTIR) spectroscopy, while their morphology was investigated by optical microscopy. Weight loss studies were carried out with R1HG-3c and R1HG-4b to investigate their water release profiles and the related kinetics, while their stability was evaluated over time both by monitoring their inversion properties to detect possible impairments of the 3D network and by PCA-assisted ATR-FTIR spectroscopy to detect possible structural changes. The flow and dynamic rheological characterization of the gels was assessed by determining their viscosity vs. shear rate, applying the Cross rheological equation to achieve the curves of shear stress vs. shear rate, and carrying out amplitude and frequency sweep experiments. Finally, their content in NH_3_^+^ groups was determined by potentiometric titrations. Due to their favorable physicochemical characteristic and the antibacterial effects of **3c** and **4b** possibly improved by the cationic R1, the pyrazole-enriched gels reported here could represent new weapons to treat severe skin and wound infections sustained by MDR bacteria of staphylococcal species.

## 1. Introduction

Pyrazole derivatives have been widely studied in the fields of medicinal chemistry because of their several biological activities including antiulcer, leishmanicidal, anti-cancer, antimalarial, antimicrobial, and cytotoxic effects [1,2,3,4,5,6]. After the discovery of the broad-spectrum antimicrobial activity of the natural pyrazole C-glycoside pyrazofurin [7], many compounds encompassing the pyrazole ring have been developed, demonstrating strong antibacterial effects [8,9,10,11]. Interestingly, some pyrazole-based compounds, known for having other pharmacological effects and already approved for clinical uses, have also shown remarkable antimicrobial properties. For instance, Celecoxib, clinically applied as a non-steroid analgesic and anti-inflammatory drug, has also evidenced significant antibacterial activity against several pathogens of clinical importance such as *Francisella tularensis*, *Francisella novicida* (MIC 16–32 μg/mL), *S. aureus*, *S. epidermidis,* and *Mycobacterium smegmatis*. Additionally, Celecoxib has been shown to increase the sensitivity of bacteria to various antibiotics including colistin, resulting in also being active against Gram-negative pathogens such as *Acinetobacter* and *Pseudomonas* [12]. Consequently, Celecoxib has inspired the synthesis of a number of its analogues endowed with more potent antibacterial activity and reduced toxicity profile. Most of the pyrazole compounds developed thus far that have shown potent antibacterial effects are -tri-substituted derivatives containing at least two phenyl rings and fluorine atoms [13]. Based on these considerations, we recently reported the synthesis, characterization, and microbiological screening of a little library of highly substituted pyrazoles, among which the -tri-substituted compounds **3c** and **4b** (Figure 1), containing two phenyl rings and fluorine atoms, demonstrated interesting antibacterial effects, especially against several multi drug resistant (MDR) clinical isolates of Gram-positive species (MICs = 32–64 µg/mL) [14]. In addition, both compounds are characterized by good drug-like and pharmacokinetic properties.

Particularly, **3c** was active against different strains of vancomycin resistant (VRE) enterococci (*E. faecium* and *E. faecalis*), methicillin resistant *S. aureus* (MRSA), and *S. epidermidis* (MRSE), while **4b** exhibited antibacterial effects specifically against staphylococci. Moreover, the data reported established that both compounds can be promising for the development of new antibacterial formulations to specifically counteract infections sustained by staphylococci, against which both ciprofloxacin and most beta-lactam drugs are no longer active. Additionally, compound **3c** can also be useful against enterococci isolates, which is difficult to inhibit with ampicillin [14]. Here, to develop topical antibacterial formulations based on **3c** and **4b**, we formulated them as hydrogels, considering that among the bactericidal/antibacterial devices, those formulated as hydrogels could effectively satisfy the urgent demand for new and successful antibacterial formulations suitable for topical administration [15].

Hydrogels consist of three-dimensional (3D) network structures obtained from synthetic and/or natural polymers capable of absorbing and retaining significant amounts of water, thus being suitable in the tissue engineering, pharmaceutical, and biomedical fields [15]. Recently, without using gelling agents, two cationic water-soluble polymers (CP1 and OP2) (Appendix A), prepared by us starting from the cationic monomers 4-ammonium methyl styrene (4-AMSTY) and 4-ammonium ethyl styrene (4-AESTY), demonstrated very low minimum inhibitory concentrations (MICs) and potent broad spectrum bactericidal effects against several clinical isolates of both Gram-positive and Gram-negative bacteria [16]. Additionally, CP1 was shown to possess unexpected gelling properties and was used to prepare a CP1-based hydrogel and a CP1OP2-based composite hydrogel, both with excellent physicochemical and rheological properties [15,17].

In particular, the strong antibacterial effects of OP2 and CP1 are based on their cationic character, which makes them like natural antimicrobial peptides (NAMPs), acting mainly through electrostatic forces and damaging the bacterial membrane, thus rapidly killing pathogens [18]. Among CP1 and OP2, the more potent bactericidal compound was the copolymer CP1, which was synthesized using the monomer 4-AMSTY (here named M1) (Figure 2) [16]. Here, to obtain a gelling agent to formulate **3c** and **4b** structurally like CP1 and potentially with similar antibacterial effects, we prepared a polystyrene-based resin (R1) using just M1 as the active monomer. Particularly, R1 was achieved by a one-step, low-cost, and scalable reverse-phase suspension copolymerization technique using M1, dimethylacrylamide (DMAA) as the copolymer, and N-(2-acryloylamino-ethyl)-acrylamide (AAEA) as a cross-linker.

R1 was characterized by several analytical techniques, which confirmed their structure and revealed a spherical morphology, micro-dimensioned particles, and high hydrophilicity. The equivalents of the NH_3_^+^ groups contained in R1, essential for a possible damaging electrostatic interaction with a bacterial surface, were estimated by the method of Gaur and Gupta [19]. Preliminary investigations proved that, upon its dispersion in an excess of water, R1 was capable of self-forming a hydrogel (namely R1HG) without using any other additive or gelling agent, which could be skin incompatible or interfere with the antibacterial effects of pyrazoles [14]. Once the favorable gelling properties of R1 were established, it was employed to formulate **3c** and **4b** as hydrogels, obtaining them with an equilibrium degree of swelling (EDS) of 765% (R1HG-3c) and 675% (R1HG-4b) and equilibrium water content (EWC) of 88% and 87%, respectively. PCA-assisted ATR-FTIR spectroscopy was used to investigate the structure of the soaked and dried gels, while their morphology was investigated by optical microscopy. Weight loss studies over time were carried out with R1HG-3c and R1HG-4b to investigate their water release profiles and the related kinetics. The stability of gels was evaluated over time by both monitoring their inversion properties to detect possible impairments of the 3D network and by PCA-assisted ATR-FTIR spectroscopy to detect possible structural changes. The flow and dynamic rheological characterization of the gels was assessed by determining their viscosity vs. shear rate, applying the Cross rheological equation and its parameters [20] to achieve the curves of shear stress vs. shear rate, and carrying out amplitude and frequency sweep experiments. Finally, the equivalents of the NH_3_^+^ groups per gram of gels was determined by potentiometric titrations.

## 2. Results and Discussion

### 2.1. An Overview on the Antibacterial Effects of **3c** and **4b**

Table 1 and Table 2 report the antibacterial effects (expressed as MICs) and the major efficiency of **3c** and **4b** with respect to some traditional antibiotics against several clinical isolates of MDR Gram-positive species [14].

Compounds were considered inactive against a specific strain when MICs higher than 128 µg/mL were observed (not bold values).

As previously observed, the data reported established that both compounds could be promising to develop new pyrazole-enriched formulations to specifically counteract infections sustained by staphylococci insensitive to ciprofloxacin and most beta lactam drugs, while compound **3c** also proved to be promising against enterococci isolates no longer inhibited by ampicillin [14].

### 2.2. Synthesis of R1

Monomer M1, whose synthesis was previously reported [16] was converted into resin R1 by reverse suspension polymerization at 35 °C. Particularly, according to slightly modified reported procedures [21,22], M1, co-monomer DMAA, and the cross-linker AAEA previously dissolved in water were suspended in a mixture of CCl_4_/hexane. SPAN 85 and APS/TMEDA were used as anti-coagulant and initiators, respectively (Figure 1a).

The polymerization was repeated several times and carried out in a cylindrical equipment such as that shown in Figure 1b. Data reported in the experimental section (ES) are those of some of the representative reactions. Importantly, containers with cylindrical geometry are able to minimize the horizontal component of the stirring motion and of the suspension, responsible for the tendency of the micro-drops to aggregate [23]. After 105 min, the polymerization was interrupted and the precipitated R1 was separated by filtration, then washed several times with isopropanol, chloroform, water, ethanol, and finally acetone. R1 was brought to constant weight at reduced pressure and stored at room temperature for subsequent sieving operations, estimation of the equivalents of NH_3_^+^ contained in the resins, and ATR-FTIR analyses. The conditions applied afforded very good conversions in the range 92–99%.

### 2.3. Sieving of R1 and Optical Microscopy

R1 was fractioned by sieving procedures, using sieves with 35–120 mesh, obtaining beads with the bulk of material equal to 77% (R1_1), 86.3% (R1_2), and 98.3% (R1_3) of the original weight, respectively. The microstructure of these particles was investigated both by optical microscopy analysis and by scanning electron microscopy (SEM). The particles appeared as microspherular beads with the bulk of material in the size range 125–250 µm. Figure 3 shows two representative images of R1, while Appendix A is a representative SEM micrograph of R1.

As observable in Figure 3, the microspheres of R1 were significantly polydisperse, as also confirmed by SEM analysis (Appendix A).

### 2.4. Estimation of the Equivalents of NH_3_^+^ Contained in R1

The equivalents of NH_2_ contained in the solid resin R1 were estimated following the method of Gaur and Gupta, which, among others, is available in the literature for determining the NH_2_ group content in insoluble matrices and is signaled out for being operator-friendly and sensitive [19]. Particularly, it is based on the labeling of the amino groups with 4-O-(4,40-dimethoxytriphenylmethyl)-butyryl residues and the quantitative determination through UV–Vis spectroscopy of the 4,40-dimethoxytriphenylmethyl cation (ε = 70,000 at 498 nm) released from the resin after treatment with HClO_4_. From the values of absorbance (A) determined at 498 nm, the NH_2_ moles for a gram of R1 were estimated as described in the ES.

According to the results, expressed as the mean of three independent determinations ± standard deviation (SD), the NH_2_ equivalents present in R1 were 10.22 ± 0.059 mmol/g.

### 2.5. Preparation of R1HG-3c and R1HG-4b

Preliminary investigations to assess the gelling properties of R1 demonstrated its high capability to absorb water and to provide a hydrogel (R1HG) with an equilibrium degree of swelling (EDS) of 900% and equilibrium water content (EWC) of 90%. In this regard, R1 appeared to be an excellent candidate to formulate **3c** and **4b** as hydrogels for potential use as a topical antibacterial formulation to treat skin or wound infection sustained by MDR bacteria of the *Staphylococcus* genus. According to the procedure described in the ES, we prepared R1HG-3c and R1HG-4b, achieving 3D networks containing the maximum amount of water that the mixtures R1/3c and R1/4b were capable of soaking up. Table 3 presents the experimental data of the preparation of R1HG-3c and R1HG-4b from 3c+R1 and 4b+R1, their EDS, the EWC of swollen gels calculated as described in the ES, as well as the details about the ingredients’ concentrations in the prepared hydrogels.

According to Table 3, the concentrations of **3c** and **4b** in the gels were 4.2 mg/mL (0.42% wt/v) (**3c**) and o 5.0 mg/mL (0.5% wt/v) (**4b**). However, the amount of **3c** and **4b** in the gels was also confirmed by UV–Vis analysis using a UV–Vis spectrophotometer (HP 8453, Hewlett Packard, Palo Alto, CA, USA) equipped with a 3 mL cuvette against the standard calibration curves of **3c** and **4b**.

Appendix A shows the obtained viscous and opaque hydrogels R1HG-3c (left side) and R1HG-4b (right side)**,** whose volume and weight corresponded to those of gels at their EDS (%), as reported in Table 3. Figure 4 shows the representative optical microphotographs of the obtained gels.

As reported in Table 3, the EWC (%) of R1HG-3c (88%) and R1HG-4b (87%) as well as their EDS (%) by volume (765% and 675%, respectively) were similar and very high. Although further experiments are necessary to confirm this early hypothesis, according to what was reported, the high porosity and remarkable capacity to absorb water could confer to the gels a high potentiality to also function as wound healing hydrogels [24,25]. The high ability to absorb water of R1HG-3c and R1HG-4b was also confirmed by the optical microscopy analysis (Figure 4). The micrographs of both hydrogels showed highly swollen brilliant spherical particles unequivocally imbued of water more than 2-fold larger than those observed when analyzing the original R1.

### 2.6. ATR-FTIR Spectra

First, the ATR-FTIR spectra were acquired on R1, on R1HG, and on the fully dried R1HG (D-R1HG). Second, we acquired the ATR-FTIR spectra of the gels prepared by dispersing R1+3c and R1+4b in water (R1HGel-3c and R1HG-4b), and of the fully dried gels (D-R1HG-3c and D-R1HG-4b) obtained by gently heating the corresponding hydrogels prepared at their EDS for about 7 h. Spectra were acquired in triplicate, and representative images of the spectra of R1 (Figure 5), R1HG-3c, and D-R1HG-3c (Figure 6a) as well as of R1HG-4b and D-R1HG-4b (Figure 6b) are reported below.

The ATR-FTIR spectrum of R1 showed weak bands of the CH stretching of methylene groups (2930 cm^−^^1^) and of NH stretching (3400 cm^−^^1^) deriving from the monomer M1, and a strong band at 1608 cm^−^^1^, typical of dimethylacrilamide derivatives, and in this case, belonging to both the co-monomer (DMAA) and the crosslinker (AAEA), thus assessing the presence of all three main ingredients in the structure of R1. No significative band was observable in the range 900–911 cm^−^^1^, thus establishing the absence of the residual monomer M1.

As expected, the ATR-FTIR spectra of the soaked R1HG-3c and R1HG-4b (blue lines in Figure 6a,b, respectively) were very simple, and showed only the typical bands of water (a large OH stretching band over 3000 cm^−^^1^ and an OH scissoring band at 1639 and 1643 cm^−^^1^), thus confirming the very high content of water in both hydrogels. Anyway, some weak bands indicating the presence of organic compounds were detectable, mainly in the spectrum of R1HG-4b at 2981, 1384, 1326, 1044, and 877 cm^−^^1^. In contrast, the ATR-FTIR of dried compounds D-R1HG-3c and D-R1HG-4b were very similar due to the presence of R1 as a major ingredient in both substances and the similar chemical structure of **3c** and **4b**, both containing the pyrazole nucleus. By comparing the spectrum of D-R1HG-3c with those of R1 and **3c**, and that of D-R1HG-4b with those of R1 and **4b**, bands deriving from R1 and typical of **3c** or **4b** were detectable (Figure 7a,b and Figure 8).

Particularly, in the spectrum of the dried D-R1HG-3c, bands deriving from R1 were observable at 2958, 2935 (CH stretching of methylene groups), 2650 cm^−^^1^, and at 1613 cm^−^^1^, (C=O stretching of the amide group of DMAA and AAEA) (Figure 7a). Additionally, bands of the main functional groups of **3c** including a weak and large band at 3286 cm^−^^1^ (OH and aromatic CH stretching), bands at 3061, 3030 cm^−^^1^, a weak band at 1671 cm^−^^1^ (C=O stretching of ureic group), and several other bands typical of **3c** at 1553, 1505, 1443, 1227, 1143, 1055, 834, and 711 cm^−^^1^ (aromatic C=C and C-N stretching) characterized the spectrum of D-R1HG-3c. It was also evidenced that the interactions of 3c-R1-H_2_O that occurred during gel formation did not affect the main functional groups of **3c** and R1. A similar scenario was observable in Figure 7b, where additional bands at 1706 and 1707 cm^−^^1^ were visible in the spectra of **4b** and D-R1HG-4b, respectively, due to the C=O stretching of the ester group of **4b**.

#### Principal Components Analysis (PCA) of the ATR-FTIR Data

First, a matrix of 30,609 variables was constructed by collecting the spectral data (wavenumbers) of **3c**, **4b**, R1, R1HG, R1HG-3c, R1HG-4b, D-R1HG-3c, D-R1HG-4b, and D-R1HG. Then, the resulting dataset was first pre-treated by autoscaling and then processed using the PCA. PCA is a chemometric tool of the multivariate analysis (MVA), capable of extracting the essential information from enormous sets of correlated variables by reducing them to a limited number of uncorrelated variables called principal components (PCs) [26]. Among the other possibilities, PCA allows one to visualize the reciprocal positions occupied by the analyzed samples in the space of the so-called score plot, where the scores are the new coordinates of samples in the new orthogonal space of the new not correlated variables (PCs). Particularly, the score plot allows one to evaluate the behavior of the samples in the space defined by the PCs, highlighting similarities and differences in their chemical composition. Here, the score plot (PC1 vs. PC2) of the nine analyzed samples is shown in Figure 9.

As observable, except for the pyrazole **4b**, all other samples were clustered into two groups separated on the PC1 based on their content of water. Particularly, the group collecting the samples with a high content of water (swollen resins) was located at negative scores, while that assembling the dried or solid materials was positioned at positive scores. Inside this latter group, R1 was sited very close to D-R1HG, while **3c** was very close to D-R1HG-3c, evidencing structural similarities among these couples of samples. Interestingly, **4b**, probably because of the ester group present in its structure, giving a distinctive band in the FTIR spectrum (1706 cm^−1^), appeared as not correlated with none of the other samples, thus resulting in being located very far from all of them. Unexpectedly, even D-R1HG-4b was very distant from **4b**, probably due to the higher concentration of R1 with respect to that of **4b**, which was not sufficient to highlight the structural similarities among them.

### 2.7. Evaluation of Stability of R1HG-3c and R1-HG-4b over Time

#### 2.7.1. PCA Assisted ATR-FTIR Spectroscopic Method: Structural Stability

ATR-FTIR spectra of R1HG-3c, R1HG-4b, and of the correspondent fully dried gels (D-R1HG-3c and D-R1HG-4b) that maintained well sealed at room temperature were acquired as described in Section 2.6 after 1, 2, 3, and 4 months. Then, to highlight possible changes in their chemical structure and composition over time, the spectral data were organized in a matrix 20 × 3401 (68,020 variables). The resulting dataset was first pre-treated by standard normal variate (SNV) normalization, which is a weighted normalization, and then processed using the PCA. The score plot of the analyzed samples (PC1 vs. PC2) is shown in Figure 10. To make the score plot clearer and avoid confusion, samples were renamed as reported in Table 4.

As shown in Figure 10, all samples (a, b, c, or d) maintained the original scores after one, two, three, and four months, and the samples a, b, c, and d analyzed over time (1, 2, 3, and 4) appeared overlapped in the score plot, thus establishing that their structural properties and pyrazole contents remained unchanged.

#### 2.7.2. Evaluation of the Inversion Properties: Stability of the 3D Network

The stability over time of the 3D network of the pyrazole-enriched hydrogels developed here was assessed as reported in the literature [27], evaluating their inversion properties after one, two, three, and four months from their first preparation, staying at room temperature. Appendix A reported in the Appendix A shows the appearance of R1HG-3c and R1HG-4b in the inverted position when just prepared, and after 1, 2, 3 and 4 months, respectively.

As observable, the inversion properties including the volumes of both gels remained unchanged over the monitoring time, and no impairments were observable, thus establishing a good stability of the hydrogels.

### 2.8. Weight Loss (Water Loss)

Appendix A shows the appearance of the fully dried D-R1HG-3c (right side) and D-R1HG-4b (left side) obtained by heating the corresponding gels R1HG-3c and R1HG-4b at 37 °C for 465 min.

As observable in Appendix A, when R1HG-3c and R1HG-4b were heat-dried, they provided apparently amorphous solids. Anyway, both optical micrographs (Figure 11a,b) and SEM images (Appendix A Appendix A) showed that both D-R1HG3c and D-R1HG-4b were made of microspherular beads like R1, having similar dimensions to those observed for R1.

Figure 12a shows the curves obtained, reporting in the graph the values of the cumulative weight loss (%) of the two gels vs. times. Figure 12b shows the kinetic models that best fit the data of the cumulative weight loss curves in Figure 12a.

As observable in Figure 12a, the weight loss was almost quantitative for both R1HG-3c and R1HG-4b (92% and 93%, respectively), and the equilibrium was reached after almost 7 h. To precisely learn the kinetics and the main mechanisms that govern the loss of water from the R1HG-3c and R1HG-4b, we fit the data of the curves in Figure 12a with a number of mathematical kinetic models including the zero-order model, first-order model, Hixson–Crowell model, Higuchi model, and Korsmeyer–Peppas model, obtaining the related dispersion graphs [28,29]. The coefficients of determination (R^2^) of the equations of the linear regressions of the obtained dispersion graphs were considered as the parameters to determine which model best fit the water loss data. The R^2^ values for the two gels are reported in Table 5 and established that the water loss from both hydrogels best fit with the first-order kinetic model (Figure 12b), with the R^2^ values being the highest ones.

As reported in the literature, first-order release kinetics state that the change in concentration of a compound released with respect to the change over time is dependent only on its residual concentration. In the present case, the release of water and weight loss over time depended only on the residual concentration of water after the heating periods [29]. First-order kinetics are described by Equation (1).
(1)$$LogQt=\left(\frac{K}{2.303}\right)\times x+LogQo$$
where *Qt* is the amount of water released on time *t*; *Qo* is the initial amount (%) of water in the gel; and *K* is the first-order constant. Accordingly, in our case, *K*/2.303 corresponded to the slopes of the linear regressions of the first-order mathematical models shown in Figure 12b, while *LogQo* was their intercepts. Consequently, the first-order constants were in both cases negative and equal to −0.0060, while the original percentage content of water (*Q_o_*) in the weighted R1HG-3c and R1HG-4b was 97% and 103%, respectively.

### 2.9. Equilibrium Swelling Rate

The swelling measurements were made at fixed times following the procedure described in ES, until the weight of the swollen resins was approximately constant.

Figure 13 shows the cumulative swelling ratio percentage curves of R1HG-3c and R1HG-4b.

As observable, the equilibrium swelling ratio (Q_equil_), which was determined at the point the hydrated resins achieved a constant weight, was reached after only 10 min by R1HG-4b and after 60 min by R1HG-3c, thus establishing for values of Q_equil_ = 1600 ± 9 (R1HG-4b) and 1560 ± 3 (R1HG-3c). In this regard, the absorption of water by R1HG-4b was very rapid, while that of R1HG-3c was similar to that reported by Baron et al. [30]. Collectively, the equilibrium swelling rates of the two gels were comparable with each other and comparable with those reported [30].

### 2.10. Potentiometric Titrations of R1HG-3c and R1HG-4b

The titration curves were obtained by graphically presenting the measured pH values vs. the aliquots of HCl 0.1N added (Figure 14, lines with round indicators). Subsequently, from the titration data, the dpH/dV values were computed and reported in the same graph vs. the corresponding volumes of HCl 0.1 N, thus obtaining the first derivative lines (D1st) of the titration curves (Figure 14, lines with square indicators), whose maxima represent the different phases of the protonation process, and how many different types of protonable nitrogen atoms exist in the samples.

As observable in Figure 14, both pyrazole-enriched hydrogels revealed a significant buffer capacity up to 1.5 mL of HCl 0.1 N added and two end titration points (see maxima of the D1st curves), evidencing a two-phase protonation profile for both samples. While the first and smallest maxima were observed for both R1HG-3c and R1HG-4b upon the addition of 2.0 mL HCl 0.1 N (maxima 3.56 and 2.38, respectively in the D1st curves), the second ones, corresponding to the most significant jumps in pH in the titration curves, were observed upon the addition of 4.0 mL HCl for R1HG-4b (maximum 4.78) and 3.5 mL HCl for R1HG-3c (maximum 4.84).

The potentiometric titrations of R1HG-3c and R1HG-4b also allowed us to titrate the equivalents of NH_2_ groups contained in the hydrogels, and to determine the NH_2_ contents per gram of gel. Table 6 presents the results related to these determinations.

According to Table 6, the NH_2_ groups contained in 1 g of hydrogels were 0.7284 mmol/g in R1HG-3c and 0.8259 mmol/g in R1HG-4b. According to data reported in Table 3 and considering that the content of cationic resin R1 in 1 g of gels is 0.070 g (R1HG-3c) and 0.082 g (R1HG-4b), the results obtained by the potentiometric titrations of gels fit perfectly with those obtained by the method of Gaur and Gupta in analyzing the dried resin R1. In fact, a NH_2_ content of 0.7284 and of 0.8259 millimoles per gram of gels should correspond to 10.41 and 10.07 millimoles per gram of resin, respectively, against the determined value of 10.22, with an error of 1.8% and 1.5% respectively.

### 2.11. Rheological Studies

To complete the physicochemical characterization of R1HG-3c and R1HG-4b, their rheological behavior was investigated. First, we determined the values of their apparent viscosity (η [Pa × s]) as a function of the applied shear rate (γ. [s^−1^]), and by plotting η values vs. γ values, we achieved the curves in Figure 15a (R1HG-3c) and Figure 15b (R1HG-4b).

Curiously, although the concentration of cationic resin R1 and pyrazoles were similar in both gels, the viscosity of R1HG-3c was significantly lower than that of R1HG-4b. Anyway, for both gels, it was observed that η decreased for small increases of γ up to values of γ < 20, while for values of γ > 20, η was practically constant and did not change significantly, at least for values of γ up to 100. Collectively, both gels are non-Newtonian fluids.

Newtonian fluids follow Newton’s law of viscosity where viscosity (η) is independent of the shear rate (γ). For Newtonian fluids, shear stress (τ) is proportional to γ, and the plot of τ vs. γ is a line with the constant slope (η) and intercept zero [31].

Differently, in non-Newtonian fluids, viscosity is shear rate dependent, and generally, non-Newtonian fluids that also exhibit a viscosity decreasing with increasing shear rate for high values of shear rate are defined as shear-thinning fluids, having values of the flow behavior index n < 1. In contrast, those fluids that for high values of γ exhibit a viscosity increasing with increasing shear rate are defined as shear thickening dilatant fluids, having n > 1. For values of γ up to 100, R1HG-3c and R1HG-4b behave as shear thinning fluids.

Furthermore, there are different types of non-Newtonian shear thinning fluids including Bingham plastic fluids, whose viscosity is constant, the graph of τ vs. γ is linear, but intercept is >0. Additionally, pseudoplastic fluids and Bingham pseudoplastic fluids exist, whose viscosity is not constant and whose graphs of τ vs. γ are not linear and have intercept zero (pseudoplastic fluids) or >zero (pseudoplastic Bigham fluids). Particularly, pseudoplastic Bigham fluids are materials that over certain values of shear rate behave as Bigham fluids, having η independent from shear rate. Therefore, to assess the actual rheological behavior of R1HG-3c and R1HG-4b, we determined the flow behavior index (n) and studied the relationship between deformation (shear rate) and shear stress. To this end, we used the Cross rheology equation [20], which can be reported in different forms. Here, we first used the Cross equation reported in the study by Xie and Jin (Equation (2)), which contains four parameters to predict the general flow behaviors of non-Newtonian fluids including *n* [20].
(2)$$\eta =\frac{\eta 0+\eta \infty \times \alpha \gamma^{n}}{\left(1+\alpha \gamma^{n}\right)}$$
where *γ* is the shear rate; η0 is the viscosity when the shear rate is close to zero; *η∞* is the viscosity when the shear rate is infinity; *n* is the flow behavior index; and *α* is the consistency index.

Following the hybrid method reported in [20], we determined the four parameters required in the Cross equation and checked the accuracy of the parameters estimated by fitting them to our experimental data. Table 7 reports the estimated parameters.

Since an ideal rheology equation relating viscosity and shear rate should provide an accurate fit for most experimental measurements over a wide range of the shear rate change [20], we assessed the goodness of the estimated parameters by determining the viscosity values for R1HG-3c and R1HG-4b, according to the Cross equation by using such parameters. From Figure 16a,b, it can be seen that the viscosity and shear rate relationships obtained (red lines) were in reasonable agreement with the experimental results (light blue lines).

Then, using the following form of the Cross equation [32] (Equation (3)) containing three parameters (consistency *α*, exponent *n*, and reference shear stress *τ*) and Equation (2), the shear stress values as a function of shear rate were obtained, solving Equation (4). The plots of shear stress vs. shear rate were obtained and are reported in Figure 17.
(3)$$\eta =\frac{\alpha }{\left(1+{\left(\frac{\alpha \gamma }{\tau }\right)}^{1-n}\right)}$$
(4)$$\left(1+{\left(\frac{\alpha \gamma }{\tau }\right)}^{1-n}\right)=\left(1+\alpha \gamma^{n}\right)$$

According to Figure 17, both gels were pseudoplastic fluids not having constant η for shear rate values <30. Over these values, their η values were constant and they assumed a plastic Bigham behavior. Their yield stress values (*τ_o_*), or tangential stress or critical strain point [33], were obtained by the intercept of the linear tendency lines in the linear tracts of the graphs and are reported in Table 7. The yield stress or yield point of a ductile material is defined as the value of the stress at which the material begins to deform plastically, passing from a reversible elastic behavior to a plastic behavior characterized by the development of irreversible deformations.

#### Frequency-Sweep Experiments

Frequency sweep experiments generate a rheological "fingerprint” of a material and are useful to elucidate the viscoelastic behavior of polymers and biomolecules. In this regard, it is important to ensure that this assay is performed within the linear viscoelastic region (LVER), which is determined by amplitude sweep experiments. Here, we first carried out dynamic strain sweep measurements to find the LVER of R1HG-3c and R1HG-4b by checking the elastic modulus (G′) and the viscous modulus (G″) at a constant frequency of 1 Hz. Generally, at small stress amplitudes, G′ does not depend on the strain, while at higher strain, a sudden downfall in the G′ occurs, indicating structural irreversible deformation and a passage from an elastic to a viscous behavior [33]. The strain value percentage, which determines the transition from the linear viscoelastic region to the viscous region, represents the dynamic yield stress defined as the minimum stress required to maintain flow [33]. From the obtained results, it was shown that the G′ of both gels was independent of strain between 0.1 and 10% in 1 Hz frequency condition. This means that below this critical strain level, the 3D network of both gels is intact and highly structured, while over such values, the network structure collapses, the material becomes progressively more fluid-like, and the modulus declines [33].

Consequently, the frequency sweep experiments were performed below the critical strain point of R1HG-3c and R1HG-4b and particularly at 1% strain to investigate the viscoelastic properties of the hydrogels’ network structure and to determine the frequency dependence of the G′ and G″ moduli. Particularly, these parameters provide information on the reversibly stored deformation energy (elastic behavior) and the irreversibly dissipated energy (viscous behavior) during one cycle, respectively [15]. As reported, hydrogels have a viscous behavior when the value of loss/viscous modulus (G″) is higher than that of the storage/elastic modulus (G′), while they have an elastic behavior if G′ > G″ [34]. Additionally, a high G′ in comparison to G″ represents strong intermolecular interactions and results from the ability to resist intermolecular slippage due to the relative strength of the loosely associated gel structure occurring in the three-dimensional network. Figure 18 shows the G′ and G″ values for R1HG-3c (blue indicators) and for R1HG-4b (red indicators), measured in the frequency range (ω) of 0.5–50 Hz and as described in the ES. As observable, for strain of 1%, the G′ values were always higher than the G″ ones for both gels and nearly independent of the angular frequency, thus suggesting that the networks of both gels have an elastic behavior like that of gels reported by Baron et al. [30].

Since the magnitude of G′ was always much larger than that of G″ for both hydrogels (Figure 18), the contribution of G″ to the magnitude of the complex modulus (G* = G′ + G″) was close to zero, and G* tends to be equal to G′. These results imply that the physical behavior of both pyrazole-based hydrogels is like that of a solid (elastic modulus).

## 3. Materials and Methods

### 3.1. Chemicals and Instruments

All reagents and solvents were from Merck (Merk Life Science S.r.l., Milan, Italy) and purified by standard procedures. AAEA was prepared by known procedures [35], while M1 was prepared as previously described [16]. Pyrazoles **3c** and **4b** were prepared as recently reported [14].

Organic solutions were dried over anhydrous magnesium sulfate and evaporated using a rotatory evaporator operating at a reduced pressure of about 10–20 mmHg. The melting ranges of solid compounds in this study were determined on a 360 D melting point device, resolution 0.1 °C (MICROTECH S.R.L., Pozzuoli, Naples, Italy). Melting points and boiling points were uncorrected. Attenuated total reflectance (ATR) Fourier transform infrared (FTIR), ^1^H, ^13^C NMR, GC-MS, GC-FID, HPLC, UV–Vis, elemental analyses, and potentiometric titrations were carried out on the same instruments as previously reported [16]. The procedures and materials to perform column chromatography and thin layer chromatography were those previously described in [16]. The optical microscopy analyses were performed using a Nikon Alphaphot-2YS2 microscope equipped with a hot stage cell (FP82HT Mettler, CH) and a 5 Mpixel live resolution digital microscopy camera (Moticam5 Motic, Canada). Image analysis and measurements were performed using Motic Images Plus 2.0ML software using a 4 × or a 10 × objective. Sieving was performed with a 2000 Basic Analytical Sieve Shaker-Retsch apparatus (Retsch Italia, Verder Scientific S.r.l., Pedrengo (BG) Italy). Finally, lyophilization and centrifugation were performed as previously described [16].

### 3.2. Synthesis of R1: General Procedure

A mixture of hexane and carbon tetrachloride (CCl_4_) was placed in a round-bottom cylindrical flanged reactor equipped with an anchor-type mechanical stirrer and nitrogen inlet, thermostated at 35 ± 0.05 °C and deoxygenated by nitrogen (N_2_) bubbling for 30 min. Meantime, a solution was obtained by dissolving in a tailed test-tube under N_2_ the monomer M1, DMAA, AAEA, and ammonium persulfate (APS) (1.8% wt/wt with respect to M1+DMAA+AAEA) in deoxygenated water distilled over KMnO_4_, and was siphoned into the reaction vessel. The density of the organic phase was adjusted by adding CCl_4_ so that the aqueous phase sank slowly when the stirring was stopped. The mechanical stirring was settled at 900 rpm, SPAN 85 dissolved in hexane was added to the mixture, and the polymerization was started. After 10 min, *N,N,N,N*-tetra-methyl-ethylene-diamine (TMEDA) was introduced and the polymerization was continued. At the end of the polymerization time, the resin was filtered, washed with 100 mL of a series of selected solvents (2-propanol, chloroform, water, absolute ethanol, chloroform, 2-propanol, and acetone in order), then dried at reduced pressure and room temperature for 16–20 h to achieve a constant weight. Table 8 presents the data of three representative polymerizations for preparing R1.

ATR-FTIR (R1) (ν, cm^−1^): 3400 (NH_3_^+^), 2930 (CH_2_), 1608 (C=O).

### 3.3. Sieving of R1

The dried resin (R1), after light grinding with a pestle to burst the fragile aggregates, was sieved using sieves with an external diameter (Ø _E_ = 10 cm) and 35–120 meshes, chosen after preliminary tests on small samples, using the analytical sieve described in Section 3.1. as a vibrating base. The morphology and microstructure of R1 were determined both by optic microscopy and scanning electron microscopy (SEM).

### 3.4. Scanning Electron Microscopy (SEM)

The microstructure of resin R1 and of the fully dried hydrogels D-R1HG-3c and D-R1HG-4b was also investigated by SEM analysis. In the performed experiments, the sample was fixed on aluminum pin stubs and sputter-coated with a gold layer of 30 mA for 1 min to improve the conductivity, and an accelerating voltage of 20 kV was used for the sample’s examination. The micrographs were recorded digitally using a DISS 5 digital image acquisition system (Point Electronic GmbH, Halle, Germany).

### 3.5. Estimation of NH_3_^+^ Content in R1 and R2

The NH_2_ content of resins R1_1 was estimated following the method of Gaur and Gupta [19]. Briefly, the resin selected as representative of R1 resins prepared (ca. 2 mg) were taken in a 2 mL Pierce reaction vial to which 4-O-(4,40-dimethoxytriphenylmethyl)-butyryl (0.25 mL) (reagent A), a catalytic amount of di-methylamino-pyridine (DMAP) (5 mg), and triethylamine (TEA) (100 µL) were added. The vial was screw-capped and gently tumbled at room temperature for 30 min. The reaction mixture was transferred to a 2-mL sintered funnel and washed successively with *N,N*-dimethylformamide (DMF) (2 × 10 mL), MeOH (2 × 10 mL), and finally with dry diethyl ether (2 × 10 mL). After the polymer support was dried under vacuum, a weighed quantity (ca. 1 mg) was placed in a 10 mL volumetric flask that was then filled up to the mark with HClO_4_ (reagent B). The released 4,40-dimethoxytriphenylmethyl cation (ε = 70,000) was estimated spectrophotometrically at 498 nm against reagent B as the blank. From the values of absorbance (*A*) determined at 498 nm, the *NH*_2_ moles for gram of *R*1 were estimated using Equation (5).
(5)$$\frac{MolesNH2}{R1(g)}=\frac{A\times V}{p\times 70,000}$$
where *V* is the volume of the mixture used for detritylations (10 mL); *p* is the weight in milligrams of the resin functionalized (1.5 mg) and subjected to detritylation.

### 3.6. Preparation of R1-HG-3c and R1-HG-4b: General Procedure

A volume of **3c** or **4b** was inserted in a graduated centrifuge tube (Ø_est_ = 14 mm, V = 10 mL), weighed, and added with about a 18.1-fold weight of R1 to obtain mixtures 3c/R1 and 4b/R1 with initial volumes Vi and initial weights Wi. Then, at room temperature and under magnetic stirring, deionized water and acetone (qb to dissolve **3c** and **4b**) were added up to a total volume of 10 mL. When a homogeneous mixture was observed, the magnetic stirrer was removed and the dispersion was sonicated at 37 °C for 15 + 15 min, and then degassed for 9 min, using a Ultrasonic Cleaner 220 V, working at a frequency of 35 kHz, timer range 1–99 min, and temperature range from 20 to 69 °C (68 to 156 °F) (VWR, Milan, Italy). The dispersion was then centrifugated at 4000 rpm for 20 min to separate the pyrazole-based gels at their maximum content of water and EDS from water in excess, which was removed. The tube was then turned upside down on filter paper to remove residual water and left for 10 min. The weights and volumes of the obtained gels corresponded to the weights and the volumes of gels at their EDS and were Wf and Vf. Accordingly, the volume of water in which **3c** or **4b** and R1 resulted finally dispersed, once the removed fraction of water not absorbed was determined. At this point, it was possible to calculate the concentration (mg/mL) of the two ingredients and of the mixtures 3c+R1 and 4b+R1. The initial volumes (*Vi*) and the final volumes (*Vf*) were used to determine the *EDS* (%) and *EWC p* (%) by Equations (6) and (7).
(6)$$EWC\left(\%\right)=\frac{Vf-Vi}{Vf}\times 100$$
(7)$$EDS\left(\%\right)=\frac{Vf-Vi}{Vi}\times 100$$

The gels were then left in the tube carefully sealed to prevent water evaporation and stored in the fridge for subsequent characterization experiments including ATR-FTIR, water loss, potentiometric titrations, and rheological studies. A fraction of the prepared gel was lyophilized to complete their characterization by optical microscopy, and to determine their swelling index.

### 3.7. ATR-FTIR Spectra

ATR-FTIR analyses were carried out on R1, on the new prepared gels (R1HG-3c and R1HG-4b), and on the fully dried gels obtained by gently heating R1HG-3c and R1HG-4b. Additionally, for comparison purposes, we acquired the spectra of the empty gel prepared with R1 (R1HG), the fully dried gel (D-R1HG) obtained by heating R1HG, and of pyrazoles **3c** and **4b**. The spectra were acquired from 4000 to 600 cm^−1^, with 1 cm^−1^ spectral resolution, co-adding 32 interferograms, with a measurement accuracy in the frequency data at each measured point of 0.01 cm^−1^ due to the internal laser reference of the instrument. Acquisitions were made in triplicate, and the spectra shown in Section 2.2 are the most representative images. The spectral data obtained were then included in a dataset matrix and were processed using the principal component analysis (PCA) by means of CAT statistical software (Chemometric Agile Tool, free down-loadable online, at: http://www.gruppochemiometria.it/index.php/software/19-download-the-r-based-chemometric-software; accessed on 15 December 2022). In particular, we arranged the FTIR data of the nine spectra in a matrix 3401 × 9 (n = 30,609) of the measurable variables. For each sample, the variables consisted of the values of transmittance (%) associated with the wavenumbers (3401) in the range 4000–600 cm^−^^1^. The spectral data in the matrix were pretreated by autoscaling.

### 3.8. Stability of R1HG-3c and R1HG-4b over Time

The structural stability of both soaked R1HG-3c and R1HG-4b and the fully dried gels (D-R1HG-3c and D-R1HG-4b) were assessed by acquiring their ATR-FTIR, as reported in Section 3.7 after one, two, three, and four months from its first preparation, staying at room temperature. Acquisitions were made in triplicate. The spectral data obtained were then included in a dataset matrix and processed using principal component analysis (PCA) by means of CAT statistical software. In particular, we arranged the FTIR data of the twenty spectra in a matrix 3401 × 20 (n = 68,020) of the measurable variables. For each sample, the variables consisted of the values of transmittance (%) associated to the wavenumbers (3401) in the range 4000–600 cm^−^^1^. The spectral data in the matrix were pretreated by the standard normal variate (SNV) normalization method and analyzed by PCA. Additionally, the stability of the 3D network of the gels was assessed by monitoring their appearance in the inverted position after one, two, three, and four months from its first preparation, staying at room temperature.

### 3.9. Weight Loss (Water Loss) Experiments

Exactly weighed samples of the swollen resins R1HG-3c (616.3 mg) and R1HG-4b (688.3 mg) were deposited in Petri dishes (PDs). The PDs were then placed in an oven under controlled temperature (37 °C) and the weight loss, meaning loss of water, was monitored as a function of time until a constant weight was reached. The cumulative weight loss percentages were determined by means of Equation (8):(8)$$WeightLoss\left(\%\right)=\frac{MQ-Mt}{MQ}\times 100$$
where *MQ* and *Mt* are the initial mass of the swollen resin and its mass after a time *t*, respectively.

### 3.10. Equilibrium Swelling Rate of Pyrazole-Based Hydrogels

The swelling measurements were carried out at room temperature by immersing 42.2 mg of fully dried R1HG-3c and 47.4 mg of fully dried R1HG-4b in deionized water (pH = 7.2–7.3) in a test tube. At intervals of time selected according to the literature [36], the sample in the test tube was centrifugated (20 min, 4000 rpm) to remove the not absorbed water, inverted on filter paper to absorb the residual water, and weighed. The cumulative swelling ratio percentage (*Q*%) as function of time was calculated from Equation (9).
(9)$$Q\left(\%\right)=\frac{WSt-WD}{WD}\times 100$$
where *WD* and *WSt* are the weights of the lyophilized gel and of the swollen gel at time *t*, respectively. The equilibrium swelling ratio (*Q*_equil_) was determined at the point (time *t*) the hydrated gels achieved a constant weight.

### 3.11. Potentiometric Titration of R1HG-3c and R1HG-4b

Potentiometric titrations were performed on R1HG-3c and R1HG-4b at room temperature and the titration curves of the gels were obtained. Exactly weighed samples of each gel (480.5 mg of R1HG-3c and 484.3 mg of R1HG-4b) were suspended in 50 mL of Milli-Q water (m-Q), and then treated under magnetic stirring with a standard 0.1 N NaOH aqueous solution (2.0 mL, pH = 10.80 for R1HG-3c and pH = 10.96 for R1HG-4b). The solutions were potentiometrically titrated under stirring by adding aliquots (0.5 mL) of HCl 0.1 N up to pH < 3, for a total volume of 5.0 [16]. Titrations were performed in triplicate and measurements were reported as mean ± SD. The titration curves shown in the Discussion section are those obtained by plotting the data obtained by carrying out the representative experiment described here.

### 3.12. Rheological Studies

The rheological properties of R1HG-3c and R1HG-4b were assayed by a continuous shear method using a Brookfield viscometer (Viscostar-R, Fungilab S.A. Torino, Italy). In particular, samples of about 2 g were subjected to shear rates ranging from 1 to 100 s^−1^. All measurements were carried out at room temperature and expressed as the mean values of five independent determinations, and the image reported in the Results and Discussion section is representative of one determination. Additionally, frequency-sweep experiments were carried out using a TA DHR-2 rheometer (TA Instruments, New Castle, DE, USA) to determine G′ (elastic/storage modulus), and G″ (loss/viscous modulus) over the angular frequency range of 0.5–50 rad/s by varying the frequency between 0.1 Hz and 10 Hz, with a fixed 1% strain, decided based upon the results from the amplitude sweep experiments performed at the fixed frequency of 1 Hz and strain (%) in the range 0.1–100. All experiments were conducted at a temperature of 37 °C and performed in triplicate.

## 4. Conclusions

In this work, new hydrogel formulations for the topical administration of two pyrazoles (**3c** and **4b**) possessing good drug-like and pharmacokinetic properties and antibacterial effects, especially against staphylococci, were developed (R1-HG-3c and R1HG-4b). As a gelling agent, we synthetized a polystyrene-based cationic resin (R1) that was proven to have high hydrophilicity, high-level porosity, and excellent swelling capabilities, thus providing pyrazole-enriched gels (R1HG-3c and R1HG-4b) with an EDS of 765% (R1HG-3c) and 675% (R1HG-4b) and EWC of 88% and 87%, respectively. The cationic character and the structural similarities of R1 with the previously reported bactericidal copolymer CP1 and resin R4 could enhance the antibacterial effects of pyrazoles due to possible electrostatic interactions with the bacterial surface. New antibacterial investigations are currently undergoing to confirm such assumptions. Here, the chemical structure of the soaked and dried gels was investigated by PCA-assisted ATR-FTIR spectroscopy, while their morphology was investigated by optical microscopy and SEM analysis. The characterization of R1HG-3c and R1HG-4b included weight loss studies, stability determinations over time, rheological experiments, and NH_3_^+^ group determinations. Collectively, the physicochemical characteristics evidenced for both R1HG-3c and R1HG-4b fully support their use as topical formulations. These findings, associated with the antibacterial effects of 3c and 4b possibly improved by the cationic R1 used as gelling agent, represent a promising basis for the development of new weapons to treat severe skin and wound infections sustained by MDR bacteria of the staphylococcal species.

## Data Availability

All data supporting the reported results are included in the present manuscript and in the associated Appendix A.

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
