# Peer review of "Hydrogel Formulations of Antibacterial Pyrazoles Using a Synthesized Polystyrene-Based Cationic Resin as a Gelling Agent"

_ijms, 2023, doi:10.3390/ijms24021109_

Round 1
Reviewer 1 Report
This article describes the synthesis and physical properties of antibacterial pyrazoles using a synthetized polystyrene-based cationic resin. Although the approach described in this manuscript are a useful for studies on antibacterial materials, it is difficult to accept it as the current manuscript for the following reasons. Reconsider after major revision is needed.
Comments
1. The important point in this paper should be the bioactivity when the compounds (3c, 4c) is polymerized. I think that this paper only shows the results of polymer property evaluation. What are the results and considerations in the examination of physiological activity?
2. I think that compound (3c, 4c) should be named a and b respectively.
3. If 4-ammonium methyl styrene (4-AMSTY) is named M1 on line 84-85, I think it should be written as M1 in the subsequent sentences. Alternatively, I think that 4-AMSTY notation is fine instead of M1. In that case, please correct the compound name in the Figures.
4. Figure 3: The letters in the scale information in the photo in b are small and difficult to understand, so I would like to make adjustments.
Author Response
This article describes the synthesis and physical properties of antibacterial pyrazoles using a synthetized polystyrene-based cationic resin. Although the approach described in this manuscript are a useful for studies on antibacterial materials, it is difficult to accept it as the current manuscript for the following reasons. Reconsider after major revision is needed.
Comments
- The important point in this paper should be the bioactivity when the compounds (3c, 4c) is polymerized. I think that this paper only shows the results of polymer property evaluation. What are the results and considerations in the examination of physiological activity?
We thank the Reviewer for his comments that enable us to clarify further the scope of our work. First, we make him kindly note that compound 4c does not exist in our manuscript, while compound 4b has been reported. Then, as reported in the title the main scope of the present work was to formulate two pyrazoles, whose significant antibacterial effects against Staphylococci, often responsible of severe skin infections have been already reported in our previous work, as hydrogels, since hydrogels are drug formulations particularly suitable for topical administration on skin. To this end, in this work we have first described the synthesis and physicochemical characterization of cationic resin R1, which we used as gelling agent to successfully formulate 3c and 4b as hydrogels, after preliminary investigation on its gelling properties. In this regard, we make kindly note to the Reviewer that, differently from his assertion, 3c and 4b were not polymerized, but only formulated in a dosage form spreadable on the skin, using R1 synthetized by us, instead of commercial gelling agents such as Carbopol. Additionally, as reported in the manuscript, we strategically customized the structure of R1 to be like those of two polymers and a cationic resin, which have demonstrated potent bactericidal properties, for its possible synergistic antibacterial effect with pyrazoles. We are confident that the pyrazoles-based hydrogels prepared here will be antibacterial agents significantly more potent than the not formulate pyrazoles 3c and 4b due to the presence of R1, but to assess the antibacterial effects of R1HG-3c and R1HG-4b was out of the scope of this first work. Differently, in this work, we wanted to provide an extensive characterization of the pyrazole-based gels to assess if they could have the physicochemical characteristic required for a topically administrable antibacterial formulation, and to verify if further biological investigations would be justifiable and rational. We think that this work is already sufficiently articulated and complete and that possible biological evaluations should have the right depth and space (that here it would be impossible to give) in a future work. Finally, as reported in the conclusions “New antibacterial investigations are currently undergoing to confirm such assumption.”
- I think that compound (3c, 4c) should be named a and b respectively.
We apologize in advance with the Reviewer, but we are forced to notify again that 4c is incorrect, while 4b is the pyrazole reported in our work. Concerning the suggestion of the Reviewer, we have used the reported names 3c and 4b for the antibacterial pyrazoles, because they are their original names reported in the previous work cited in the present ones (Ref. 14). We think that using different names could generate confusion and could break the continuity of the research line. Then, we ask the Reviewer to accept our choice.
- If 4-ammonium methyl styrene (4-AMSTY) is named M1 on line 84-85, I think it should be written as M1 in the subsequent sentences. Alternatively, I think that 4-AMSTY notation is fine instead of M1. In that case, please correct the compound name in the Figures.
As suggested by the Reviewer, we have standardized the manuscript by using always M1 to name the monomer 4-ammonium methyl styrene. Please, see lines 93, 95 and 153.
- Figure 3: The letters in the scale information in the photo in b are small and difficult to understand, so I would like to make adjustments.
As asked the due adjustments have been made. Now the letters in the scale information are clearly visible and easy to read.
Reviewer 2 Report
The introduction is sometimes difficult to read, as it is not clearly explained what the reference polymer CP1 and OP2 are. I suggest that the author improve the second half of the introduction so that it is easier to be understood also by chemist that are not familiar with the previous works of the same authors.
Table 3, the explanation of the meaning of Vi, Wi, Vf is missing, although explained in the experimental section
There are 11 self-citations. While I understand that they are relevant for the topic of the paper, I believe that they should be reduced to a maximum of 5.
What the authors call "maximum swelling capability percentage (S%)" is more commonly known in the hydrogel literature as Equilibrium degree of swelling (EDS). "Porosity percentage (P%)" is instead known as Equilibrium water content (EWC). I suggest that the authors employ this more common nomenclature.
The microarchitecture of the hydrogel should be evaluated not only with optical microscopy, but also with ESEM or SEM.
The rheological analysis focus only on shear viscosity, and no data are presented on oscillatory rheology (G', G"). These data are crucial to understand the behavior of any hydrogel and I suggest that the authors also perform amplitude sweep and frequency sweep experiments.
In Table 7, the Cross model is employed to study the yeld stress of the hydrogel. However, the author report a negative infinite shear viscosity. The authors should repeat the data analysis imposing that infinite shear viscosity is always positive. Also, in the description of Equation 2 the authors use the Greek letter "mu", while in the equation 2 they use the letter "eta".
The quality of the PCA pictures should be improved and aligned to all the other pictures.
Author Response
The introduction is sometimes difficult to read, as it is not clearly explained what the reference polymer CP1 and OP2 are. I suggest that the author improve the second half of the introduction so that it is easier to be understood also by chemist that are not familiar with the previous works of the same authors.
We thank the Reviewer for his rational comment. As asked, we have improved the part signaled by the Reviewer, thus making clearer the roles of CP1 and OP2. Additionally, we inserted a new image (Figure S1) in the Supplementary Materials (SM) showing the chemical structure of CP1 and OP2. Please, see lines 76-96 and Figure S1 in SM.
Table 3, the explanation of the meaning of Vi, Wi, Vf is missing, although explained in the experimental section
The Reviewer is right. Explanations concerning Vi, Wi and Vf have been included in the footnotes of Table 3. Please, see lines 212-213.
There are 11 self-citations. While I understand that they are relevant for the topic of the paper, I believe that they should be reduced to a maximum of 5.
The Reviewer is right, 11 self-citations are excessive. So, they were reduced. Now, the first-name citations are 5, as asked.
What the authors call "maximum swelling capability percentage (S%)" is more commonly known in the hydrogel literature as Equilibrium degree of swelling (EDS). "Porosity percentage (P%)" is instead known as Equilibrium water content (EWC). I suggest that the authors employ this more common nomenclature.
We thank a lot the Reviewer for his useful suggestion. As asked, the nomenclature used by us has been replaced with that suggested by the Reviewer. Please, see lines 16-17, 107-109, 199-201, 208, 222-223, 227-228, 242, 626-627, 630, 634-635, 722-723 and equations 6 and 7.
The microarchitecture of the hydrogel should be evaluated not only with optical microscopy, but also with ESEM or SEM.
We thank the Reviewer for his suggestion with which we agree. Fortunately, the technician in charge of the SEM analysis was available despite the Christmas period and it was possible to acquire the SEM images of R1 and of fully dried D-R1HG-3c and D-R1HG-4b. The obtained micrographs have been inserted in the SM so as not to burden the main text with further Figures. Please, see Figure S2, S6 and S7 in SM and the SEM experimental part at the lines 592-598.
The rheological analysis focus only on shear viscosity, and no data are presented on oscillatory rheology (G', G"). These data are crucial to understand the behavior of any hydrogel and I suggest that the authors also perform amplitude sweep and frequency sweep experiments.
We agree with the Reviewer. So, we completed the rheological analyses performing amplitude and frequency sweep experiments and reporting in the revised version of manuscript the experimental procedure (lines 709-715), the results, the related discussion (lines 502-541) and the new Figure 18.
In Table 7, the Cross model is employed to study the yeld stress of the hydrogel. However, the author report a negative infinite shear viscosity. The authors should repeat the data analysis imposing that infinite shear viscosity is always positive. Also, in the description of Equation 2 the authors use the Greek letter "mu", while in the equation 2 they use the letter "eta".
We appreciated the observation of the Reviewer, and we checked carefully our data and calculations which resulted corrected. The values of viscosity at infinite shear rate were obtained by the graphical method proposed in ref 20, where such values correspond to the intercept of the equations of the linear tendency line of the regression graphs obtained plotting the values of viscosity vs. . In this regard, the values of the intercepts obtained were those reported in Table 7. We retain incorrect to impose different data. In addition, the correctness of parameters determined was confirmed by determining the viscosity values for R1HG-3c and R1HG-4b according to the Cross equation, using such parameters. From Figure 16a and 16b, it can be seen that the viscosity and shear rate relationships obtained were in reasonably agreement with the experimental results. Concerning the second issue signalled by the Reviewer, it has been corrected. Please, see line 468.
The quality of the PCA pictures should be improved and aligned to all the other pictures.
We apologize in advance with the Reviewer, but unfortunately the PCA pictures have been provided as such by CAT software that in the version downloadable for free (as the one used by us) does not allow to personally modify or rework the graphic style of the images. We therefore kindly ask the reviewer to approve the PCA images that we have provided.
Round 2
Reviewer 1 Report
Thank you for your comments and revisions. I was satisfied with the revised manuscript.